# Self-reported abortion experiences in Ethiopia and Uganda, new evidence from cross-sectional community-based surveys

**Margaret Giorgio**[1]*, **Fredrick Makumbi**[2], **Simon Peter Sebina Kibira**[2], **Solomon Shiferaw**[3], **Assefa Seme**[3], **Suzanne O. Bell**[4], **Elizabeth Sully**[1]

1 Guttmacher Institute, New York, NY, United States of America, 2 School of Public Health, College of Health Sciences, Makerere University, Kampala, Uganda, 3 School of Public Health, Addis Ababa University, Addis Ababa, Ethiopia, 4 Johns Hopkins Bloomberg School of Public Health, Department of Population, Family, and Reproductive Health, Baltimore, MD, United States of America

* mgiorgio@guttmacher.org

**Data Availability Statement:** The community-based survey dataset from Uganda analyzed during the current study are openly available from

## Abstract

Unsafe abortion is a major contributor to maternal morbidity and mortality. To gain insight into the ways in which abortion restrictions and stigma may shape reproductive health outcomes, we present self-reported data on abortions in Ethiopia and Uganda and compare these findings across the two varying legal contexts. W investigate differences in sociodemographic characteristics by whether or not a woman self-reported an abortion, and we describe the characteristics of women's most recent self-reported abortion. In Ethiopia only, we classified abortions as being either safe, less safe, or least safe. Finally, we estimate minimum one-year induced abortion incidence rates using the Network Scale-Up Method (NSUM). We find that women who self-reported abortions were more commonly older, formerly married, or had any children compared to women who did not report an abortion. While three-quarters of women in both settings accessed their abortion in a health facility, women in Ethiopia more commonly used public facilities as compared to in Uganda (23.0% vs 12.6%). In Ethiopia, 62.4% of self-reported abortions were classified as safe, and treated complications were more commonly reported among least and less safe abortions compared to safe abortions (21.4% and 23.1% vs. 12.4%, respectively). Self-reported postabortion complications were more common in Uganda (37.2% vs 16.0%). The NSUM estimate for the minimum one-year abortion incidence rate was 4.7 per 1000 in Ethiopia (95% CI 3.9–5.6) and 19.4 per 1000 in Uganda (95% C 16.2–22.8). The frequency of abortions and low levels of contraception use at the time women became pregnant suggest a need for increased investments in family planning services in both settings. Further, it is likely that the broadly accessible nature of abortion in Ethiopia has made abortions safer and less likely to result in complications in Ethiopia as compared to Uganda.

Performance Monitoring and Accountability at https://www.pma2020.org/request-access-to-datasets. The community-based survey dataset from Ethiopia analyzed during the current study are available openly available at https://pma.ipums.org/pma/index.shtml

**Funding:** The study was made possible by grants given to the Guttmacher Institute from The David and Lucile Packard Foundation, an anonymous foundation, and The William and Flora Hewlett Foundation. The views expressed are those of the authors and do not necessarily reflect the positions and policies of the donors. The funders had no role in study design, data collection and analysis, decision to publish, or preparation of the manuscript.

**Competing interests:** The authors have declared that no competing interests exist.

## Background

Induced abortion is common in East Africa. Model-based estimates for 2015–2019 indicate that approximately 35 per 1,000 women of reproductive age have an induced abortion each year in the region [1]. While abortions rarely result in medical complications when conducted in accordance with internationally accepted standards [2], only 24% of abortions that occur in East Africa are classified as safe (i.e., performed by a trained provider using a recommended method) [3]. As such, unsafe abortion is a major contributor to maternal morbidity and mortality in the region; in 2019, the annual abortion case fatality rate in sub-Saharan Africa was 185 maternal deaths per 100,000 abortions, which translates to approximately 15,000 preventable maternal deaths each year [4].

Ethiopia and Uganda represent very different legal and abortion stigma environments. Abortion remains highly restrictive in Uganda and is only legal in order to save a woman's life [2, 5]. Conversely, Ethiopia expanded its abortion law in 2005 to make abortions available in many cases, including for women who become pregnant due to rape or incest, who are younger than 18, who have physical or mental disabilities, whose life or physical health would be at risk if the pregnancy continued, who are physically or mentally unprepared for childbirth, or in the case of fetal impairment [6]. While abortion-related stigma is high in both countries, a recent systematic review classified women in Ethiopia as experiencing higher levels of abortion stigma in comparison to women in Uganda [7].

Comparing estimates of abortion incidence and safety across different legal contexts can highlight the ways in which abortion restrictions and stigma influence women's health outcomes. For example, a recent study used community-based surveys of women to compare estimates of abortion safety from two countries where abortion is highly restricted (Nigeria and Cote d'Ivoire) to a context where abortion is broadly legally available (Rajasthan, India) [8]. The results of this work showed that abortions classified as less safe (those that used non-clinically recommended methods and/or providers) were far more common in the legally restricted settings [8]. This type of comparative work can also highlight gaps in current programs aimed at improving the provision of family planning, post-abortion care, or safe abortion care services.

Understanding the state of abortion in a particular context is challenging due to limitations of currently available methodologies. In Ethiopia and Uganda, most available national-level evidence on abortion was generated from studies that used the Abortion Incidence Complications Method (AICM)conducted in 2014 in Ethiopia and 2013 in Uganda [9, 10]. The AICM uses health facility data and knowledge from health professionals and key informants to estimate abortion incidence and provide information about abortion safety. As such, little is known about abortions that occur entirely outside of the formal healthcare system. This is problematic, as out-of-facility abortions accounted for approximately half of all induced abortions in Ethiopia in 2014 [9], and the legally restrictive environment in Uganda means that abortions that occur outside the formal healthcare system are common. Further, the increasing availability of misoprostol through informal sources in sub-Saharan Africa has likely diminished the reliability the AICM estimates over time [11, 12].

An alternative method for gathering information on abortion is asking women directly about their experiences in surveys. This data is notoriously biased; asking survey respondents directly about their abortion often results in underreporting [13–15], and it is likely that respondents who self-report abortions in a community-based survey differ systematically from those who do not. However, the changing landscape of abortion access and the limited availability of alternative methods have made direct surveys of reproductive aged women a critical source for understanding the safety of abortion. Self-reported abortion data can also be used to provide information on the characteristics of women who are able to access abortions in a

given context. While women in all socio-demographic groups obtain abortions, it is important to understand how these factors may be associated with differential access to abortion services. For example, previous research has shown that educational attainment, wealth status, and urban residence are all associated with increased rates of induced abortion [16]. As such, research on the characteristics of women who obtain abortions is critical for identifying and understanding inequities in sexual and reproductive health care and outcomes.

This study aims to provide updated descriptive data on induced abortion in Ethiopia and Uganda using self-reported data from community-based surveys of women. We first describe information from women's direct reports of their abortion experiences to better understand the characteristics of women who self-report an abortion in each country and the circumstances under which their abortions occurred. We also compare these findings across countries to gain insight into the ways in which abortion restrictions and stigma may be shaping reproductive health outcomes. We also estimate one-year induced abortion incidence rates using the Network Scale-Up Method (NSUM), which is a social network based approach that relies on third party reporting [17]. Given existing biases in the application of the NSUM to measure abortion [18], we present these estimates as likely minimum rates of the incidence of induced abortion in Ethiopia and Uganda.

## Methods

### Ethics statement

Ethical approval for this study was provided by the Institutional Review Boards of the Guttmacher Institute, Johns Hopkins Bloomberg School of Public Health, Makerere University, and Addis Ababa University, as well as the Uganda National Council for Science and Technology. Informed consent was obtained from all adult respondents. Both parental/guardian consent and minor assent was obtained for all respondents under age 18.

### Data sources and sample

This analysis utilizes data from household-based surveys of women of reproductive age in Ethiopia and Uganda. Data in Ethiopia were collected in March 2020 and come from a panel study designed as a follow-up to the 2018 Performance Monitoring for Action (PMA) survey [19]. Data in Uganda come from the 2019 round of the PMA platform [20]. Sampling procedures varied by survey and country and are documented in detail in S1 Text. In brief, the sample in Ethiopia was only representative of 6 regions (Addis Ababa, Afar, Amhara, Oromia, SNNPR, Tigray), where 90% of the population live [21]. Of the 6,306 women eligible for the survey, 4,909 were successfully interviewed (response rate = 78%). We weighted these data to account for differential selection and loss to follow-up from the 2018 PMA Ethiopia female survey in order to make the 2020 sample nationally representative. To construct the weights, we regressed age, education, marital status, residence, and wealth from the 2018 survey on participation in the 2020 survey. We then took the inverse of the predicted probability of participation in the follow-up survey from the regression and multiplied it by the selection probability from the 2018 survey to produce a weight for each 2020 respondent. In Uganda, the sampling procedure for the female survey resulted in a nationally representative sample of women aged 15–49. A total of 4,767 women were sampled, and 4,481 were successfully interviewed (response rate = 94%). We use the PMA constructed weights for national representativeness. All surveys were conducted face-to-face on Android smartphones using Open Data Kit (ODK) software [22].

## Measures

We collected data on respondent characteristics in both countries, including age, educational attainment (no education, primary, secondary, post-secondary), marital status (currently married or cohabiting, formerly married, never married), urban/rural residence, and parity. To measure induced abortion among respondents, women were asked if they have ever done anything to end a pregnancy. Respondents who indicated "yes, and I succeeded" were coded as ever having had an abortion. Respondents were then asked for the month and year of their most recent abortion.

Information was also collected about the circumstances under which the respondent's most recent abortion occurred. Women reported whether they were using contraception at the time they became pregnant, and if so what method. We collapsed method type into five categories: permanent method (male/female sterilization), long-acting reversible methods (LARCs) (IUD, implants), short-acting modern methods (pill, injectables, emergency contraception, male condom, female condom, diaphragm, foam/jelly, LAM, standard days/cycle beads) (Diaphragm and foam/jelly for Uganda only), tradition methods (rhythm method, withdrawal, other traditional methods), and no method use. Respondents also reported where they went to end their pregnancy, which included public health facilities/services, private health facilities/services, pharmacies/drug shops, and non-health system sources (i.e., churches, friends, traditional healers, the woman herself). If the respondent reported going to more than one place, she was asked to report the final place that she went to end the pregnancy. In Ethiopia only, women reported the method they used to induce their abortion, which we categorized into surgical procedures, medication abortion (misoprostol alone or in combination with mifepristone), and other methods (pills other than misoprostol/mifepristone, injections, traditional medicines, insertion of materials into the vagina, alcohol, or other methods.) If a woman indicated she used more than one method, she was asked to report the first and last method used.

We assessed abortion safety using several indicators. According to a framework proposed by Ganatra et al. (2017), *safe abortions* are conducted using a WHO-recommended method (medical abortion, vacuum aspiration, or dilatation and evacuation) by a trained provider, *less safe abortions* are those that only meet one of criteria of safe abortions, and *least safe abortions* are those provided by untrained individuals using unrecommended methods [3]. Due to limitations in our available data, we used proxy measures for whether the abortion was performed by a trained provider and whether the abortion method was recommended by the WHO. First, we assumed that all facility-based abortions (public or private) were performed by a trained provider, and all out of facility abortions were performed by an untrained provider. Second, if a woman reported that she had a surgical procedure, we assumed that it was either vacuum aspiration or dilatation and evacuation (All self-reported surgical abortions in Ethiopia were also reported to have occurred within a health facility.). Finally, we classified women has having a recommended medical abortion if the respondent reported using misoprostol alone or in combination with mifepristone; if a woman reporting using pills of unknown type, we assumed they were not misoprostol and/or mifepristone. Applying these proxy measure to the Ganatra framework, we categorized abortions in Ethiopia as being "safe" if a woman reported she had a surgical or medical abortion that was performed in either a public or private health facility. "Less safe" abortions were defined by only meeting one of those criteria, and "least safe" abortions met neither. We were not able to create a similar abortion safety measure for Uganda due to the lack of data on abortion methods in the 2019 PMA survey. In order to gain more insight into the safety of abortion in each country, we also measured whether the respondent experienced a health complication from her most recent abortion that she later

sought care for at a health facility. While not a direct measure of abortion safety, levels of reported abortion complications provide insight into the underlying safety of abortions.

## Analysis

First, we present the distribution of socioeconomic characteristics of each sample. Next, we used chi-squared tests and t-tests to investigate bivariate differences in women's self-report of ever having an abortion for categorical and continuous characteristics, respectively, using a p-value cutoff of ≤0.05 to determine statistical significance. Among women self-reporting an induced abortion, we present descriptive statistics for characteristics of the respondent's only/most recent abortion. In Ethiopia only, we present the distribution of self-reported abortions based on this study's abortion safety measure, and we assessed whether there were differences in the frequency of treated abortion complications by abortion safety.

To estimate abortion incidence rates, we first used women's self-reported abortion information; one-year incidence estimates were calculated as the number of women who reported an induced abortion in the past 12 months divided by the number of women in the sample. Estimates were then multiplied by 1,000 to get the incidence per 1,000 women and weighted using the individual sample weights that account for the complex survey design.

We also used the Network Scale-Up Method (NSUM) to estimate minimum abortion incidence rates in each country. The NSUM was first fielded in the 2018 rounds of the PMA female surveys in Ethiopia and Uganda, and again the 2019/2020 female surveys described in this paper. The methodologies and analyses were nearly identical across the two survey rounds in each country, and a detailed summary is described elsewhere [18]. In brief, we first estimated the sizes of respondents' social networks using the 'known population' approach, where each respondent is asked to report the number of women she knows who have a certain characteristic for which the size of that population in the country is known (i.e., the number of women she knows who are teachers) [23, 24]. Populations of known sizes were selected using the most recent Demographic and Health Surveys for Ethiopia and Uganda [25, 26]. After collecting data from respondent on several populations of known sizes, we estimated respondent's personal network size using an established formula [17, 23, 24], and conducted an internal validity check to test how well the method performed (see S1 Text for additional details).

Each respondent was asked how many women they know who have ever successfully induced an abortion in the past 12 months. We calculated the one-year induced abortion incidence rate by dividing the sum of the number of social network members that respondents reported having done anything to successfully induce an abortion in the 12 months prior to the survey by the sum of respondents' social network sizes, which is then multiplied by 1,000 [17, 23, 24]. Further details on the NSUM estimation procedure can be found in the S1 Text.

While the NSUM has several strengths that make it appealing for the measurement of induced abortion incidence [18, 27], one limitation is the difficultly in measuring and accounting for "transmission bias", which is the likelihood that respondents will have imperfect and/or incomplete knowledge of all induced abortions that occur within her social network [28, 29]. Attempts to measure transmission bias in these surveys were unsuccessful [18]. As such, we do not attempt to adjust the NSUM induced abortion incidence estimates for transmission bias and instead present them as likely *minimum* abortion rates in each setting.

In order to put the study's incidence estimates in context, we present them in comparison to the most recent estimates of abortion incidence in each country, which were generated as part of a global study that used Bayesian modeling to estimate country-specific abortion incidence rates for the 2015–2019 time period [30]. It is important to note that this new model

relies heavily on the 2014/2013 Ethiopia and Uganda AICM estimates as inputs for the estimation process [30].

All analyses were performed using Stata version 16.0 (StataCorp LP, College Station, TX).

## Results

Overall, women in Ethiopia were slightly older and had less years of education than women in Uganda (Table 1). Distributions of urban residence, marital status, and parity were similar across the two contexts. Few women self-reported ever having had an abortion in each country (Ethiopia: 3.7%, n = 181; Uganda: 4.3%, n = 191). In both countries, we observed statistically significant differences in several socio-demographic characteristics between women who self-reported an induced abortion and those who did not; women who self-reported ever having had an induced abortion were older (Ethiopia: mean of 35.3 vs. 29.9, p<0.001; Uganda: 32.9 vs. 28.1, p<0.001) (Table 1). In addition, larger proportions of women who self-reported abortions were formerly married (Ethiopia: 22.6% vs. 10.5%, p<0.001; Uganda: 29.2% vs. 13.2%, p<0.001), or had any children (Ethiopia: 88.0% vs. 71.9%, p = 0012; Uganda: 88.5% vs. 74.9%, p = 0.014). While larger proportions of women who self-reported an abortion had a secondary education or greater in both countries, this difference was only statistically significant in Ethiopia (Ethiopia: 36.0% vs. 24.2%; Uganda: 43.8% vs. 34.9%). Similarly, larger proportions of women who self-reported an abortion lived in urban areas compared to rural areas, although

**Table 1. Sociodemographic characteristics of women in Ethiopia (2020) and Uganda (2019), overall and by self-reported abortion status, weighted*.**

| | Ethiopia | | | | | | Uganda | | | | | |
| --- | --- | --- | --- | --- | --- | --- | --- | --- | --- | --- | --- | --- |
| | Overall | | Self-reported abortion | | No self-reported abortion | | p-value | Overall | | Self-reported abortion | | No self-reported abortion | | p-value |
| | (N = 4909) | | (N = 181) | | (N = 4728) | | | (N = 4481) | | (N = 191) | | (N = 4290) | | |
| Age, mean (SD)• | 30.1 | 9.6 | 35.3 | 8.2 | 29.9 | 9.6 | <0.001 | 28.3 | 9.4 | 32.9 | 8.2 | 28.1 | 9.5 | <0.001 |
| Education, %(n)‡ ⬦ | | | | | | | <0.001 | | | | | | | 0.228 |
| Never | 47.9 | 1556 | 31.3 | 27 | 39.2 | 1529 | | 8.9 | 548 | 7.2 | 14 | 9.0 | 534 | |
| Primary | 35.1 | 1572 | 32.7 | 57 | 36.6 | 1515 | | 55.8 | 2420 | 49.0 | 87 | 56.1 | 2333 | |
| Secondary or higher± | 17.0 | 1779 | 36.0 | 97 | 16.5 | 1682 | | 35.3 | 1521 | 43.8 | 90 | 34.9 | 1422 | |
| Residence, %(n)‡ ⬦ | | | | | | | <0.001 | | | | | | | 0.055 |
| Urban | 23.6 | 2548 | 62.4 | 156 | 22.7 | 2392 | | 23.2 | 1202 | 32.9 | 92 | 22.8 | 1110 | |
| Rural | 76.4 | 2361 | 37.6 | 25 | 77.3 | 2336 | | 76.8 | 3279 | 67.1 | 99 | 77.2 | 3180 | |
| Union/marital status, %(n)‡ ⬦ | | | | | | | <0.001 | | | | | | | <0.001 |
| Married/cohabiting | 67.1 | 3187 | 72.3 | 125 | 67.0 | 3062 | | 63.6 | 2801 | 62.5 | 118 | 63.6 | 2683 | |
| Formerly married | 10.8 | 606 | 22.6 | 42 | 10.5 | 564 | | 13.8 | 626 | 29.2 | 54 | 13.2 | 572 | |
| Never married | 22.1 | 1116 | 5.1 | 14 | 22.5 | 1102 | | 22.6 | 1053 | 8.3 | 19 | 23.2 | 1034 | |
| Parity, %(n)‡ ⬦ | | | | | | | 0.012 | | | | | | | 0.014 |
| No children | 27.8 | 1366 | 12.0 | 23 | 28.1 | 1343 | | 24.5 | 1137 | 11.5 | 23 | 25.1 | 1114 | |
| 1–2 | 25.9 | 1432 | 36.9 | 77 | 25.7 | 1355 | | 31.9 | 1377 | 33.3 | 59 | 31.8 | 1318 | |
| 3–5 | 26.4 | 1324 | 29.9 | 61 | 26.3 | 1263 | | 25.2 | 1132 | 31.7 | 64 | 25.0 | 1068 | |
| 6+ | 19.9 | 787 | 21.2 | 20 | 19.9 | 767 | | 18.3 | 801 | 23.5 | 41 | 18.1 | 760 | |

*Table presents weighted estimates with unweighted Ns

‡ Ns may not sum to total N due to missing data. Valid percents shown.

± Secondary or higher includes: Secondary, Technical, Higher (Ethiopia); 'O' Level, 'A' Level, Tertiary, University (Uganda)

• T-tests for statistical significance used

⬦ Chi-squared tests for significance used

this difference was only marginally significant in Uganda (Ethiopia: 62.4% vs. 22.7%, p<0.001; Uganda: 32.9% vs. 22.8%, p = 0.0552),

Among women self-reporting ever having had an induced abortion, 32.1% in Ethiopia and 19.4% in Uganda reported using a contraceptive method at the time she become pregnant (Table 2). Most commonly, women reported using a short acting method (Ethiopia = 82.4%, Uganda = 69.4%). Approximately three-quarters of women accessed their abortion in a health facility setting (Ethiopia = 74.2%, Uganda = 72.6%). In both countries, private facilities were more commonly reported than public facilities (Ethiopia: private = 51.1%, public = 23.0%; Uganda: private = 60.0%, public = 12.6%). Unsurprisingly given the legal status of abortion in each country, a larger proportion of women in Ethiopia reported accessing their abortion at a public facility compared to women in Uganda (23.0% vs 12.6%). The proportion of women who reported experiencing an abortion complication that was treated in a health facility was more than twice as high in Uganda (37.2%) than in Ethiopia (16.0%).

In Ethiopia, approximately 44.7% of women reported using medication abortion methods to end their pregnancy, 29.6% reported surgical, and 25.7% reporting using another type of method. Approximately two-thirds (62.4%) of self-reported abortions in Ethiopia are classified as safe. Of the remaining abortions, 14.4% were less safe and 23.2% were least safe. Abortions that resulted in a treated complication were more commonly reported among women with least and less safe abortions compared to safe abortions (21.4% and 23.1% vs. 12.4%, respectively; Fig 1).

**Table 2. Characteristics of self-reported abortions in the last 12 months in Ethiopia and Uganda\*.**

| | Ethiopia (N = 181) | | Uganda (N = 191) | |
|---|---|---|---|---|
| | n | % | n | % |
| Respondent reported contraception use at the time she became pregnant | 52 | 32.1 | 37 | 19.4 |
| Method‡ | | | | |
| LARC | 1 | 2.0 | 3 | 8.3 |
| short acting | 42 | 82.4 | 25 | 69.4 |
| traditional method | 8 | 15.7 | 8 | 22.2 |
| Where woman accessed the abortion | | | | |
| Health facility | 132 | 74.1 | 138 | 72.6 |
| *Public* | *41* | *23.0* | *24* | *12.6* |
| *Private* | *91* | *51.1* | *114* | *60.0* |
| Pharmacy/drug shop | 9 | 5.1 | 8 | 4.2 |
| Non-health system/other | 37 | 20.8 | 44 | 23.2 |
| Experience a complication for which the woman sought care in a health center. | 29 | 16.0 | 71 | 37.2 |
| Abortion method | | | | |
| Surgical | 53 | 29.3 | n/a | n/a |
| Medication± | 80 | 44.2 | n/a | n/a |
| Other | 46 | 25.4 | n/a | n/a |
| Abortion safety classification | | | | |
| Safe | 113 | 62.4 | n/a | n/a |
| Less safe | 26 | 14.4 | n/a | n/a |
| Least safe | 42 | 23.2 | n/a | n/a |

\*Table displays unweighted data

‡Among women who reported using method at the time she became pregnant

±Medication abortions refer to abortions where the woman reported using misoprostol alone or in combination with mifepristone

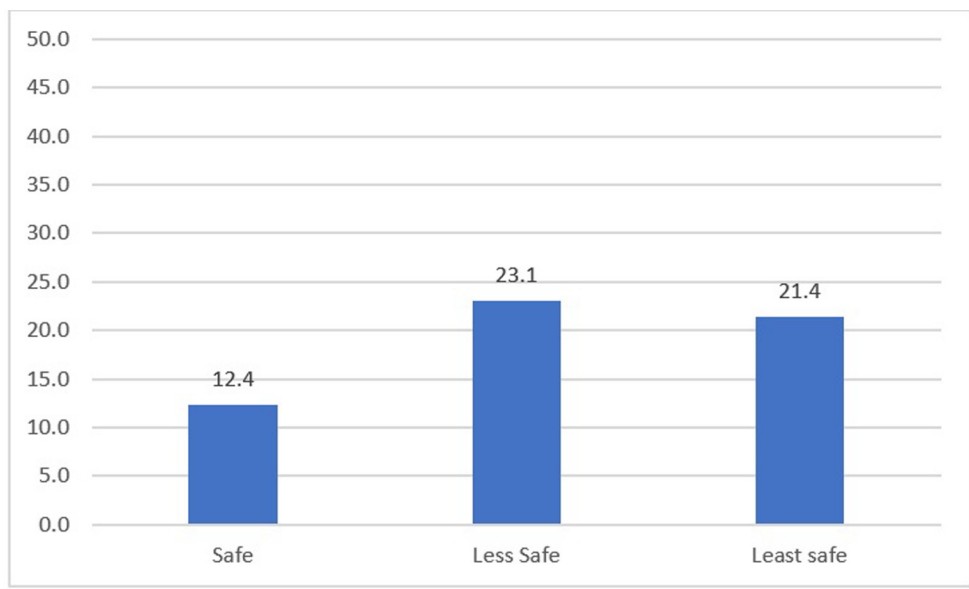

**Fig 1. Frequency of treated complications by abortion safety categories in Ethiopia, 2020.**

Fig 2 displays estimates of the annual abortion incidence rate in each country. The abortion incidence rate estimated using Bayesian modeling was 24 per 1,000 women aged 15–49 in Ethiopia (95% UI 17–35) and 43 per 1,000 in Uganda (95% UI 29–60) [30]. Induced abortion incidence estimates generated from this study's self-reported abortions were low in comparison, at only 2.6 per 1000 in Ethiopia (95% CI 2.0–3.4) and 4.5 per 1000 in Uganda (95% CI 3.5–5.7). Internal validity checks indicated that the NSUM performed well in estimating social network sizes in both countries (see Fig A and Table A in S1 Text). The unadjusted NSUM estimate for the minimum one-year abortion incidence rate was 4.7 per 1000 women aged 15–49 in Ethiopia (95% CI 3.9–5.6) and 19.4 per 1000 (95% C 16.2–22.8) in Uganda.

## Discussion

The results of this study provide valuable new information about induced abortions and the circumstances under which they occur in Ethiopia and Uganda. Making conclusions about the true distribution of the characteristics of women who have abortions or abortion safety from these data is difficult due to known abortion-related reporting biases in community-based surveys. However, these data still provide important insights into women's abortion experiences in Ethiopia and Uganda. Further, our comparison of the results from two differing legal contexts provides evidence for how abortion restricts may influence abortion incidence and/or safety in a given setting.

We found that women who self-reported an induced abortion differed systematically from women who did not; women who self-reported abortions in both countries were older and had more children, and were also less likely to be married or live in a rural area. In addition, larger proportions of women who self-reported abortions were in the highest education level category, although this difference was only statistically significant in Ethiopia. There is evidence to suggest that some of these differences may reflect the true distribution of abortion experiences and access in the underlying populations in each country. For example, the average age difference by self-reporting status is likely a function of more reproductive years and opportunities to become pregnant. It is also well documented that abortion services are more

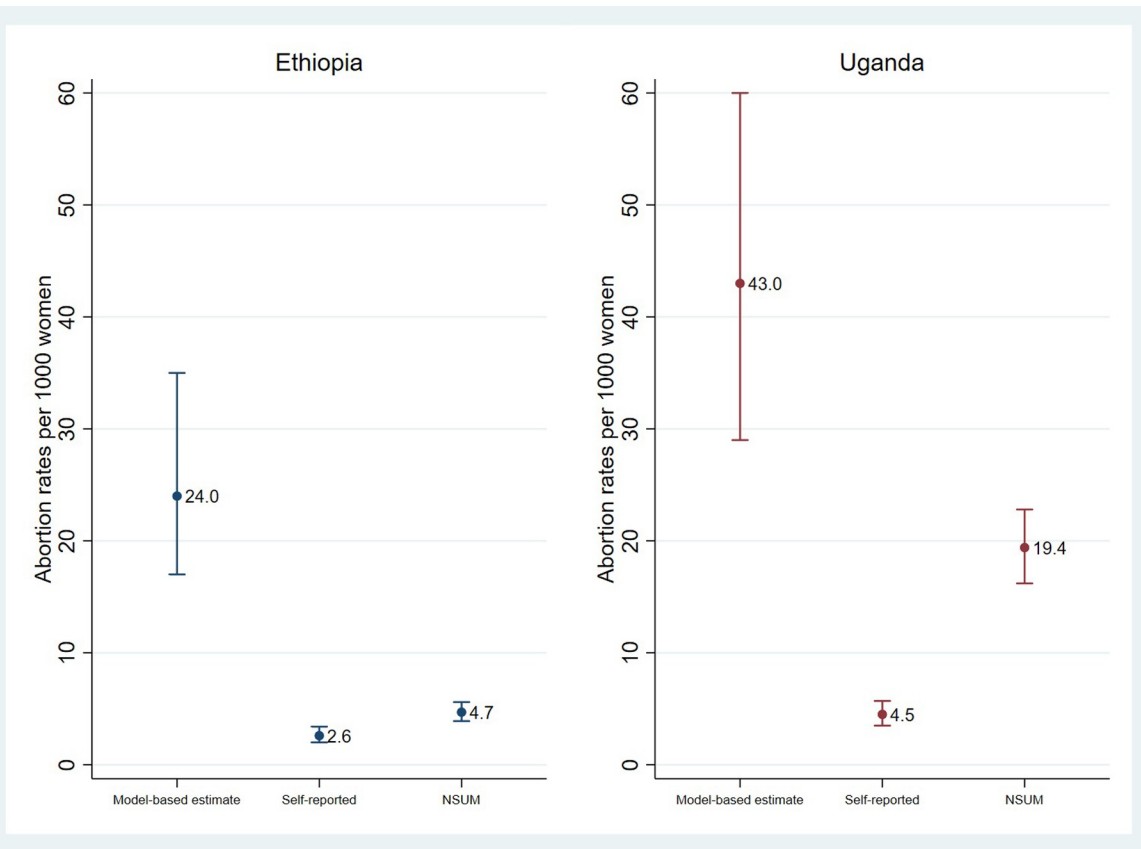

**Fig 2. One-year abortion incidence estimates for Ethiopia and Uganda.**

widely available in urban as opposed to rural areas [4], and the observed differences by urban/rural status may reflect this differential access to services. Further, unintended pregnancy rates are generally lower among married women, which may explain marital status differences by self-reported abortion status. Similarly, the higher educational attainment of women who self-reported an abortion may be a reflection of greater access to abortion services or differences in fertility preferences associated with educational attainment.

While the characteristics of women who self-reported an abortion differ in ways that we would expect, the results from this study may still be subject to reporting biases. Consistent with previous work [14, 15], our analyses suggest that women underreported their own experiences with abortion in each survey, as evidenced by the difference between the self-reported abortion incidence rates and the minimum NSUM estimated rates. As such, differences in self-reported abortion may also reflect reporting biases due to abortion-related stigma. For example, the greater proportion of women self-reporting an abortion who had any children and/or were currently/previously married could reflect cultural norms around pre-marital sex and early childbearing such that women who have induced abortions before they have children or are married may be less likely to report these abortions. Young people, especially adolescents, are more constrained in matters of sexual and reproductive health by social norms than older people [31], and young unmarried women may withhold information about their abortions. Given the low rates of abortion self-reporting and the observed differences in sociodemographic characteristics by reporting status, our estimates describing the circumstances under which women's abortions occurred are almost certainly biased and not representative of

all abortions in Ethiopia or Uganda. Despite this limitation, it is still useful to examine the abortion experiences of women's self-reported abortions, especially when comparing experiences across two contexts with different legal restrictions on induced abortions.

In both Ethiopia and Uganda, most women reported non-use of modern contraception at the time she became pregnant prior to her only/most recent abortion, indicating that unmet need is high among women at the greatest risk of abortion. Unmet need for modern contraception use, as defined as the gap between women's reproductive intentions and their contraceptive behavior, has been shown to be common in Ethiopia and Uganda in recent years [32]. While Ethiopia and Uganda have made great gains in increasing access to modern contraceptives in recent years [25, 33–35], increasing investments in family services in both country may assist in reducing the rate of unintended pregnancies and abortion.

Comparing the characteristics of induced abortions across legal contexts can reveal important insights into the ways the legal context of induced abortion can impact abortion access and safety. The majority of women in both countries reported accessing abortion services from a health facility setting, but larger proportions of women in Ethiopia reported accessing their abortion through a public health facility than women in Uganda. This difference is likely due Ethiopia's more liberal abortion law, which allows for greater access to safe abortion care in the public sector. However, private health care providers were the most commonly reported abortion provider in both countries. This finding may be partially explained by the biased sample of women who self-reported abortions; these women were more highly educated and wealthy, likely resulting in a greater ability to pay for abortion services in the private sector. This finding is also indicative of the important role that the private sector plays in abortion provision in each country. Previous work has argued that choice within the abortion service landscape, both in terms of available providers and methods, is essential for advancing sexual and reproductive health and rights [36, 37]. In the context of Ethiopia, where abortion is already legally accessible, future research should investigate why some women prefer to use private services. The results of this work could be used to help improve service delivery in the public sector, which may promote more equitable access to safe abortion services. However, given the length of time it may take to understand this phenomenon, it may also be important to consider investments in safe abortion care services in the private sector as a complement to the government's continued focus on strengthening the infrastructure of public sector abortion services [6, 9, 38]. This will help ensure greater access to quality abortion care services that meets women's preferences and needs.

Overall, approximately two-thirds of the self-reported induced abortions in Ethiopia were classified as "safe" in this study. This is much higher than the regional estimate for East Africa of only 24% [3]. Some of this difference may be explained by the legal context in Ethiopia, where safe abortion services are widely accessible. The 2014 induced abortion incidence study found that the proportion of abortions that occurred outside of health facilities declined from 2008 to 2014 [9], and it is possible that access to safe abortion services has continued to increase over time. However, our safety estimate may not reflect the true distribution of the safety of induced abortions in the underlying population. We do not have detailed information on the exact procedures and abortion methods, and it is possible that respondents did not accurately report the methods that were used. Further, our assumption that all facility-based abortions were performed by trained providers, and that all non-facility based abortions were performed by untrained providers, may also be incorrect. Misclassification for either of these indicators (methods used or provider training levels) may have biased our safety estimates. Finally, even if all abortion information was accurately reported, women who self-reported abortions in our survey may differ systematically from those who did not, which may have biased our results.

The lack of information on abortion providers and methods in the Uganda data limits our ability to draw conclusions about the safety of the self-reported abortions from that survey. However, several scholars have identified limitations to the Ganatra et al. (2017) safety definition, highlighting that its focus on procedural aspects of abortion (type of provider and methods used) does not take into account the actual outcomes of abortion [39, 40]. This is particularly true with the rise of informal access to medication abortion. While these abortions would be classified as a "less safe" using the procedural safety definition, several studies have demonstrated that many or most women can safely induce abortions using medication abortion despite the fact they were obtained from untrained/informal providers [41–43]. The recently updated guidelines from the WHO acknowledge this point, noting that with accurate information women may be able to safely manage their abortion with misoprostol and/or mifepristone themselves [44].

Focusing on abortion the available data on abortion outcomes, there is evidence to suggest that the self-reported abortions in Uganda were less safe than those reported in Ethiopia; in Uganda, women more commonly reported that their abortion resulted in a complication that was treated in a health facility as compared to Ethiopia, which may be evidence of more severe negative health outcomes of abortions. It is important to interpret these self-reported data with caution, as some women may have conceptualized normal bleeding after a medication abortion as a complication in need of treatment. That said, recent work has document that, on a global level, abortions are more likely to be unsafe in countries with more restrictive abortion laws [3]. The legal context in Ethiopia allows for safe abortion care to be supported and regulated by the government, meaning that formal providers likely have the necessary training to provide safe abortion care. It is possible that this context also allows more informal providers to have access to information on safe abortion care and practices in Ethiopia. Together, this environment is likely partially responsible for the lower self-reported rate of complications in Ethiopia in this study.

Our analyses provide new minimum one-year abortion incidence estimates for Ethiopia and Uganda. The true abortion incidence rate for each country is higher than the ones presented in this paper, as these estimates do not account for transmission error (aka the inability of respondents to report on the abortions that are unknown to them). As a theoretical example, if women are aware of approximately one-third of the abortions that occur within their social networks in Uganda, the transmission bias adjusted induced abortion incidence estimate would be 63.9 per 1000 women. To date, accurate estimates of the magnitude of abortion-related transmission bias are not available for Ethiopia or Uganda. Previous efforts to measure abortion visibility within social networks have failed [18, 45], and future research is needed to improve the measurement of transmission bias so that social network-based methods for estimating abortion incidence can be appropriately adjusted for abortion visibility.

Despite the limitations of the NSUM incidence rates presented in this study, they are still valuable to policy makers and researchers. First, while the recent Bayesian model-based country-specific abortion incidence estimates are an important contribution to the global study of unintended pregnancy and abortion, their usefulness is highly dependent on the quality of the underlying data inputs. For example, the model relies heavily on previously estimated abortion incidence rates when available [30]. In the case of Ethiopia and Uganda, these were the AICM estimates from the 2014/2013 studies, and evidence of this reliance can be seen in the similarities between the AICM and model-based incidence rates in each country (Ethiopia: 28 vs. 24 per 1,000, respectively; Uganda: 39 vs. 43 per 1,000, respectively) [9, 10, 30]. The biases that exist in the AICM estimates are not fully accounted for or corrected in the Bayesian model, and improvements to the model-based estimates will only be achieved by providing updated, high-quality estimates of abortion incidence, as well as other model inputs. In the context of

this study, our internal validation tests indicate that the NSUM performed well in estimating social network sizes, and these minimum abortion incidence estimates can be used as additional inputs to improve the global model-based estimates of abortion incidence and unintended pregnancy. The continued improvement of these key sexual and reproductive health indicators can assist key stakeholders track trends overtime and better plan for contraceptive service provision targets [1, 30].

## Conclusions

Taken together, this study's data on the characteristics of women who self-report abortions, the safety of those reported abortions, and minimum abortion incidence rates provides important insights into how to improve policies to prevent unintended pregnancies and improve access to safe abortion care. Despite the identified biases, these results can be used to monitor key abortion-related indicators over time, including abortion methods and safety. Differences in the characteristics of women who self-report abortions suggest that access to abortion services remains limited for some groups. The frequency of abortions and reported contraception use at the time women became pregnant suggest a need for increased investments in family planning services in both settings. Further, it is likely that the broadly accessible nature of abortion in Ethiopia has made abortions safer in Ethiopia as compared to Uganda. Increasing access to safe abortion care in Uganda will likely improve outcomes for women.

## Supporting information

**S1 Text. Supplemental materials: Self-reported abortion experiences in Ethiopia and Uganda, new evidence from cross-sectional community-based surveys.** Sampling Strategies for Each Survey; Additional information on the Network Scale-Up Method analyses.
(DOCX)

**S1 Checklist. Inclusivity in global research.**
(DOCX)

## Acknowledgments

We would like to thank the entire field team for administering the surveys and the respondents for participating. We thank Lily Ha and Doris Chiu for research assistance and Ann Biddlecom for reviewing an earlier version of this manuscript.

## Author Contributions

**Conceptualization:** Margaret Giorgio, Assefa Seme, Suzanne O. Bell, Elizabeth Sully.

**Data curation:** Solomon Shiferaw, Assefa Seme.

**Formal analysis:** Margaret Giorgio, Elizabeth Sully.

**Methodology:** Margaret Giorgio, Suzanne O. Bell, Elizabeth Sully.

**Project administration:** Solomon Shiferaw, Assefa Seme.

**Supervision:** Margaret Giorgio, Assefa Seme, Suzanne O. Bell.

**Validation:** Elizabeth Sully.

**Writing – original draft:** Margaret Giorgio.

**Writing – review & editing:** Solomon Shiferaw, Assefa Seme, Suzanne O. Bell, Elizabeth Sully.

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
