## [Decision Letter · Decision Letter 0]

2 May 2023

PGPH-D-23-00098

Self-reported abortion experiences in Ethiopia and Uganda, new evidence from cross-sectional community-based surveys

Dear Dr. Giorgio,

Thank you for submitting your manuscript to PLOS Global Public Health. After careful consideration, we feel that it has merit but does not fully meet PLOS Global Public Health’s publication criteria as it currently stands. Therefore, we invite you to submit a revised version of the manuscript that addresses the points raised during the review process.

We look forward to receiving your revised manuscript.

Kind regards,

Jianhong Zhou

Staff Editor

Journal Requirements:

1. Please include a complete copy of PLOS’ questionnaire on inclusivity in global research in your revised manuscript. Our policy for research in this area aims to improve transparency in the reporting of research performed outside of researchers’ own country or community. The policy applies to researchers who have travelled to a different country to conduct research, research with Indigenous populations or their lands, and research on cultural artefacts. The questionnaire can also be requested at the journal’s discretion for any other submissions, even if these conditions are not met.  Please find more information on the policy and a link to download a blank copy of the questionnaire here: https://journals.plos.org/plosone/s/best-practices-in-research-reporting. Please upload a completed version of your questionnaire as Supporting Information when you resubmit your manuscript.

2. Please provide separate figure files in .tif or .eps format.

Additional Editor Comments (if provided):

Reviewers' comments:

Reviewer's Responses to Questions

**Comments to the Author**

1. Does this manuscript meet PLOS Global Public Health’s publication criteria? Is the manuscript technically sound, and do the data support the conclusions? The manuscript must describe methodologically and ethically rigorous research with conclusions that are appropriately drawn based on the data presented.

Reviewer #1: Yes

Reviewer #2: Yes

2. Has the statistical analysis been performed appropriately and rigorously?

Reviewer #1: I don't know

Reviewer #2: Yes

3. Have the authors made all data underlying the findings in their manuscript fully available (please refer to the Data Availability Statement at the start of the manuscript PDF file)?

Reviewer #1: Yes

Reviewer #2: Yes

4. Is the manuscript presented in an intelligible fashion and written in standard English?

Reviewer #1: Yes

Reviewer #2: Yes

5. Review Comments to the Author

Reviewer #1: The authors describe how legal restrictions and stigma related to induced abortion in each setting influence the experience of women who have had induced abortions, as well as the safety of their induced abortion in a comparative study between two sub-Saharan African countries. This study is very important because induced abortion remains a leading cause of maternal mortality in the region, and information on induced abortion outside of health facilities remains very limited.

The approach is also original because there are still very few studies using network methods to contribute to more information on the incidence and safety of the many abortion cases that are not captured by conventional approaches to assessment in health facilities; the comparison between two countries also provides insight into how the difference in legal context may influence behaviour in accessing care as well as the level of safety of induced abortions.

In addition, the authors have well described the limitations of the methodological approaches used, such as reporting bias, transmission bias, and difficulties in weighting.

We have made some comments to help improve the quality of this manuscript.

The manuscript is written in a clear and understandable language, however there are still some mistakes that persist in the text, as well as abbreviations that are cited without having been spelled beforehand. the correction of all these typos would improve the quality of the final manuscript.

231-233: ‘Next, we use chi-squared tests and t-tests to investigate bivariate differences in these characteristics based on whether women self-reported ever having an abortion.’

The authors should specify in which case they used one or the other test, and the conditions of applicability of the statistical model (p-value, confidence interval). This will help to better understand the rigor of their analysis and the accuracy of their interpretation of the results.

286-288: ‘In both countries, there were statistically significant differences in all socio-demographic characteristics between women who self-reported an induced abortion and those who did not’

As noted above, the lack of precision on the significance level does not allow us to verify the authors' interpretation of the results. Indeed, in Table 1, the p-values for the variables 'Education' and 'Residence' for Uganda leave us in doubt.

290-294: ‘In addition, larger proportions of women who self-reported abortion had a secondary education or greater (Ethiopia: 36.0% vs. 24.2%; Uganda: 43.8% vs. 34.9%), lived in urban areas (Ethiopia: 62.4% vs. 22.7%; Uganda: 32.9% vs. 22.8%), were formerly married (Ethiopia: 22.6% vs. 10.5%; Uganda: 29.2% vs. 13.2%), or had any children (Ethiopia: 88.0% vs. 71.9%; Uganda:88.5% vs. 74.9%).’

The authors should interpret the results as presented in Table 1. Indeed, the variable "education" is categorized into 4 modalities and not 2, which confuses the interpretation of the authors for the reader.

326-329: ‘Of the remaining abortions, 14.4% were less safe and 23.2% were least safe. Abortions that resulted in a treated complication were more commonly reported among least and less safe abortions compared to safe abortions (21.4% and 23.1% vs. 12.4%, respectively; Figure 1).’

The authors should discuss these results a little more, as they highlight the quality of care received in the health facilities, in a context where the practice is legal. In addition to strengthening the structural capacity of health facilities, the availability of management protocols, equipment and inputs, and qualified and motivated staff should be ensured.

352-354 ‘We found that women who self-reported an induced abortion differed systematically from women who did not, with women who self-reported abortions in both countries being older, more educated, with more children, as well as less likely to be married or live in a rural area.’

As noted above, the limited information provided by the authors makes it impossible to agree with them on these conclusions, and it would be risky to make recommendations based on such an interpretation.

408-412 ‘In the context of Ethiopia, where abortion is already legally accessible,

it may be important for donors and service providers to consider investments in safe abortion

care services in the private sector as a complement to the government’s continued focus on

strengthening the infrastructure of public sector abortion services. (6,9,38) This will help ensure greater access to quality abortion care services that meets women’s preferences and needs.’

We are concerned that donors’ emphasis on private health facilities will contribute to inequities and inequalities in access to safe abortion services. It would be interesting to investigate why, despite the broad legal context, women prefer to use private services and try to solve these problems for a wider and more equitable access

Reviewer #2: Reviewer comment/ suggestions for the authors

- Self-reported data have inherent limitations and potential sources for bias. For this, the authors are not to be blamed. In fact they should be congratulated for recognizing, acknowledging and discussing them.

- The study provides a wealth of information. It may be important to highlight findings that are most relevant to health policymakers, and let them stand out clearly. Two findings deserve particular emphasis and are relevant when two countries with different abortion legal contexts are compared and it would be good to have then highlighted in the abstract, discussion and conclusions sections:

-Self-reported treated post-abortion complications were more than twice as high in Uganda where abortion is restricted (37.2% ) compared to 16.0% in Ethiopia.

-Estimated abortion incidence was much higher in Uganda where it is legally restricted. The minimum one year abortion incidence rate was estimated as 4.7 per 1000 in Ethiopia and 19.4 in Uganda

- Procedure safety (as defined for public health purposes) is no more important than actual patient safety as reported complications. Patient safety data deserve more emphasis and detail.

- If data were available in Ethiopia before and after the abortion law was expanded, it would be good to mention in the discussion.

- Reference 4 is not complete. It is one of the good Guttmacher reports.

6. PLOS authors have the option to publish the peer review history of their article (what does this mean?). If published, this will include your full peer review and any attached files.

**Do you want your identity to be public for this peer review?** For information about this choice, including consent withdrawal, please see our Privacy Policy.

Reviewer #1: No

Reviewer #2: No

---

## [Decision Letter · Decision Letter 1]

8 Aug 2023

Self-reported abortion experiences in Ethiopia and Uganda, new evidence from cross-sectional community-based surveys

PGPH-D-23-00098R1

Dear Dr Giorgio,

We are pleased to inform you that your manuscript 'Self-reported abortion experiences in Ethiopia and Uganda, new evidence from cross-sectional community-based surveys' has been provisionally accepted for publication in PLOS Global Public Health.

Best regards,

Olivia Miu-yung Ngan

Academic Editor

Reviewer Comments (if any, and for reference):

Reviewer's Responses to Questions

**Comments to the Author**

1. If the authors have adequately addressed your comments raised in a previous round of review and you feel that this manuscript is now acceptable for publication, you may indicate that here to bypass the “Comments to the Author” section, enter your conflict of interest statement in the “Confidential to Editor” section, and submit your "Accept" recommendation.

Reviewer #1: All comments have been addressed

Reviewer #2: All comments have been addressed

2. Does this manuscript meet PLOS Global Public Health’s publication criteria? Is the manuscript technically sound, and do the data support the conclusions? The manuscript must describe methodologically and ethically rigorous research with conclusions that are appropriately drawn based on the data presented.

Reviewer #1: Yes

Reviewer #2: Yes

3. Has the statistical analysis been performed appropriately and rigorously?

Reviewer #1: Yes

Reviewer #2: Yes

4. Have the authors made all data underlying the findings in their manuscript fully available (please refer to the Data Availability Statement at the start of the manuscript PDF file)?

Reviewer #1: Yes

Reviewer #2: Yes

5. Is the manuscript presented in an intelligible fashion and written in standard English?

Reviewer #1: Yes

Reviewer #2: Yes

6. Review Comments to the Author

Reviewer #1: (No Response)

Reviewer #2: (No Response)

7. PLOS authors have the option to publish the peer review history of their article (what does this mean?). If published, this will include your full peer review and any attached files.

**Do you want your identity to be public for this peer review?** For information about this choice, including consent withdrawal, please see our Privacy Policy.

Reviewer #1: No

Reviewer #2: No
